# Current use of attention-deficit hyperactivity disorder (ADHD) medications and clinical characteristics of child and adolescent psychiatric outpatients prescribed multiple ADHD medications in Japan

**Yoshinori Sasaki**[1,2], **Noa Tsujii**[3], **Shouko Sasaki**[2], **Hikaru Sunakawa**[2], **Yusuke Toguchi**[2], **Syuuichi Tanase**[2], **Kiyoshi Saito**[2], **Rena Shinohara**[2], **Toshinari Kurokouchi**[2], **Kaori Sugimoto**[2], **Kotoe Itagaki**[4], **Yukino Yoshida**[5], **Saori Namekata**[4], **Momoka Takahashi**[5], **Ikuhiro Harada**[6], **Yuuki Hakosima**[2], **Kumi Inazaki**[2], **Yuta Yoshimura**[2], **Yuki Mizumoto**[2], **Takayuki Okada**[1], **Masahide Usami**[2,4,5]*

1 Department of Psychiatry and Behavioral Science, Tokyo Medical and Dental University Graduate School, Tokyo, Japan, 2 Department of Child and Adolescent Psychiatry, Kohnodai Hospital, National Center for Global Health and Medicine, Chiba, Japan, 3 Department of Neuropsychiatry, Kindai University Faculty of Medicine, Osaka, Japan, 4 Department of Clinical Psychology, Kohnodai Hospital, National Center for Global Health and Medicine, Chiba, Japan, 5 Clinical Center of Children's Mental Health, Kohnodai Hospital, National Center for Global Health and Medicine, Chiba, Japan, 6 Department of Social Worker, Kohnodai Hospital, National Center for Global Health and Medicine, Chiba, Japan

* usami.masahide@hospk.ncgm.go.jp, usami.masahide@gmail.com, usami.masahide@mac.com

**Data Availability Statement:** Due to ethical restrictions upon the dataset by the Ethical Committee of the National Center for Global Health

## Abstract

### Background/aim

Patients with attention-deficit hyperactivity disorder (ADHD) manifest symptoms of hyperactivity, impulsivity, and/or inattention. ADHD medications available in Japan are limited compared with those in Western countries. Prescribing status has not been sufficiently evaluated in clinical settings in Japan. This study investigated the current use of ADHD medications and characteristics of patients who received multiple ADHD medications in a clinical setting in Japan.

### Methods

Study participants were those who visited the Department of Child and Adolescent Psychiatry, Kohnodai Hospital between April 2015 and March 2020. We investigated patients who received osmotic-controlled release oral delivery system methylphenidate, atomoxetine, or guanfacine. A retrospective case–control design was used to evaluate the characteristics of patients who received multiple ADHD medications. Patients who were given three ADHD medications were defined as the case group. Randomly sampled sex- and age-matched patients diagnosed with ADHD were defined as the control group. We compared data for child-to-parent violence, antisocial behavior, suicide attempt or self-harm, abuse history,

and Medicine (Tokyo, Japan), the data underlying this study may not be made publicly available. The anonymized dataset is available upon request to interested and qualified researchers who meet the requirements for access to confidential data. Data access requests can be sent to the Ethical Committee of the National Center for Global Health and Medicine through the email address rinrijm@hosp.ncgm.go.jp.

**Funding:** This study was supported in part by Grants-in-Aid for Research from the National Center for Global Health and Medicine (20A3001 and 21A1012). The funder played no role in the study design, data collection and analysis, decision to publish, or preparation of the manuscript.

**Competing interests:** The authors have declared that no competing interests exist.

refusal to attend school, and two psychological rating scales (the ADHD-Rating Scale and Tokyo Autistic Behavior Scale).

## Results

Among the 878 patients who were prescribed any ADHD medications, 43 (4.9%) received three ADHD medications. Logistic regression revealed that children with severe ADHD symptoms, autistic characteristics, or tendency of child-to-parent violence were more likely to have been prescribed three medications during their treatment.

## Conclusions

Our findings suggest the approach to prevent the use of multiple ADHD medications. A prospective study to investigate the causality between prescribing status and clinical characteristics is warranted.

## Introduction

Patients with attention-deficit hyperactivity disorder (ADHD) manifest symptoms of hyperactivity, impulsivity, and/or inattention in childhood. The symptoms affect cognitive, academic, behavioral, emotional, and social functioning [1]. The estimated prevalence of ADHD was found to be 7.2% in children and adolescents aged <19 years [2]. ADHD is one of the most common disorders in child and adolescent psychiatry.

Treatment of ADHD involves pharmacotherapy, psychosocial treatment, or a combination of both [3]. Current clinical guidelines for ADHD recommend pharmacotherapy, with or without behavioral intervention, as the first-line treatment for school children (≥6 years) and adolescents with uncomplicated ADHD [3]. Several types of medications are available to treat ADHD symptoms, including stimulants (e.g., methylphenidate, dexmethylphenidate, dextroamphetamine, dextroamphetamine-amphetamine, and lisdexamfetamine), norepinephrine reuptake inhibitors (e.g., atomoxetine), and alpha-2-adrenergic agonists (e.g., guanfacine) [4–6]. However, as of December 2019, only three ADHD medications have been approved in Japan: osmotic-controlled release oral delivery system (OROS) methylphenidate, atomoxetine, and guanfacine. The number of ADHD medications available in Japan is insufficient compared with that in other countries.

The most recent Japanese guideline was published in 2016, which listed only OROS methylphenidate and atomoxetine as ADHD medications [7]. The most recent Japanese expert consensus on pharmacotherapy for ADHD was published in 2015, which also evaluated only OROS methylphenidate and atomoxetine [8]. As the fourth medication after guanfacine, the long-acting stimulant lisdexamfetamine has been covered by Japanese insurance since December 2019. Lisdexamfetamine is defined as a stimulant raw material by the Amphetamines Control Law, and therefore, there is a note stating the following: "Use only if other ADHD medications are ineffective." However, the prescribing status of ADHD medications has not been thoroughly examined in a clinical setting.

This study investigated the current use of ADHD medications and characteristics of patients who received multiple ADHD medications in a clinical setting in Japan. It is inappropriate to simultaneously use all three medications for the potential risks of side effects. However, it seems reasonable to use three medications sequentially in certain conditions. The

comorbidity of ADHD and autism spectrum disorder (ASD) is approximately 30%–50% [9, 10]. Methylphenidate was effective in treating hyperactivity associated with ASD, but the magnitude of the response was less than that noted in typical ADHD [11]. We hypothesized that patients who received multiple ADHD medications tend to have more autistic symptoms than those who did not. ADHD frequently occurs simultaneously with other neurodevelopmental conditions, such as ASD.

## Methods

### Study design and setting

This study used data from the Registry Study of Child and Adolescent Mental Health in Japan (http://www.ncgmkohnodai.go.jp/subject/100/optout.html). The study participants included patients who visited the Department of Child and Adolescent Psychiatry, Kohnodai Hospital, National Center for Global Health and Medicine between April 2015 and March 2020. We investigated patients with ADHD who received OROS methylphenidate, atomoxetine, or guanfacine. In Japan, only patients with ADHD diagnosis can receive ADHD medication, regardless of whether it is a primary or secondary diagnosis, and with or without comorbidities. We examined the proportion of patients who had been prescribed for ≥180 days to determine the response rate of each drug treatment based on previous studies [12, 13]. Furthermore, a retrospective case–control design was used to evaluate the clinical characteristics of patients resistant to ADHD medications. Patients who received all three medications, including OROS methylphenidate, atomoxetine, and guanfacine, were defined as the case group. The patients in the case group had ADHD as their diagnosis. Conversely, randomly sampled sex- and age-matched patients diagnosed with ADHD at initial consultation during the same period were defined as the control group. Therefore, this study aimed to investigate differences between patients diagnosed with ADHD and patients who received multiple ADHD medications. For the two-group comparison, we compared ADHD diagnosed children with a use of no more than 2 ADHD drugs (i.e., tried zero or one or two drugs but not all three) versus ADHD diagnosed children who were treated with all three ADHD medications. In line with the case–control design, we compared data for child-to-parent violence, antisocial behavior, suicide attempt or self-harm, abuse history, refusal to attend school, and two psychological rating scales, the ADHD-Rating Scale (ADHD-RS) [14] and the Tokyo Autistic Behavior Scale (TABS) [15].

The study protocol was approved by the ethical committee of the National Center for Global Health and Medicine (Tokyo, Japan) and conducted in accordance with the tenets of the Declaration of Helsinki. The Ethical Guidelines for Medical and Health Research Involving Human Subjects of Japan state that "Observational study only using past clinical records and not human tissue samples does not necessarily require informed consent from study participants, but researchers must publish information on the implementation of the study, including the purpose of the study." The purpose, methods, and analyses of the study and details of how to refuse participation were posted in the hospital's outpatient clinic. In addition, the study data were anonymized because patient correspondence was unnecessary throughout the study period. Information from this study may contain potentially identifiable patient information, and data sharing is restricted by the Ethical Committee of the National Center for Global Health and Medicine based on the Ethical Guidelines for Medical and Health Research Involving Human Subjects of Japan. However, the data could be accessed by contacting the Ethical Committee of the National Center for Global Health and Medicine through the email address rinrijm@hosp.ncgm.go.jp.

## Recruitment and participants

This study included patients aged ≤15 years at the first consultation who visited our department between April 2015 and March 2020. Patients aged <1 year also came to the clinic, but teenagers made up majority of patients. Most participants resided in Ichikawa City and the surrounding areas, which has a population of approximately 492,393 (as calculated in August 2020).

Psychologists and psychiatrists together established the initial interview forms, which included demographic and clinical characteristics. Consultation included the developmental history from caretakers, child observations, and school record reviews. TABS and ADHD-RS were then constructed. Psychiatrists specializing in child and adolescent psychiatry evaluated clinical findings, including psychological rating scales and diagnosed all patients according to the Diagnostic and Statistical Manual of Mental Disorders, Fifth Edition [1]. Meetings to discuss the psychopathology, symptoms, diagnosis, and possibility of abuse were conducted as needed. There were few patients with moderate-to-severe intellectual disability according to the DSM-5, organic brain disease, drug-induced psychiatric disease, traumatic brain injury, and genetic syndromes because they were referred to other medical institutions when they call for an appointment in this study.

In this study, the TABS was filled out by caretakers and then psychiatrists checked the items in a medical interview. The TABS score indicated the strength of autistic characteristics, and the ADHS-RS indicated the strength of ADHD symptoms as well. Child-to-parent violence meant physical violence against parents living together or damage to property. The in-charge psychiatrists interviewed patients and parents individually, as needed, to evaluate for child-to-parent violence. After hearing both sides of the story, psychiatrists decided whether the patients' behavior was indicative of child-to-parent violence. We considered patients as having "antisocial behavior" when patients engaged in illegal activities, such as smoking, drinking, illegal drug use, shoplifting, or causing injury to others. In Japan, smoking and drinking are legal after the age of 20 years. "Abuse experience" included sexual abuse, physical abuse, psychological abuse, and neglect. Our department worked with local child protection services. Hence, information about abused children was often available before the initial consultation. In addition, the in-charge psychiatrists interviewed parents and patients separately to identify abused children as needed. These efforts were made to immediately identify abused children and to prevent missed abused children. Refusal to attend school was defined as absence from school for ≥30 days owing to any psychological, emotional, physical, or social reasons [16].

## Measurements

**ADHD-RS.**   ADHD-RS is an 18-item tool used by a child's caretaker to assess ADHD symptoms [14]. Takayanagi et al. standardized the Japanese version of the ADHD-RS, which has two factors: hyperactivity/impulsiveness and inattention. Each item is scored from 0 to 3, with a score of 54 indicating the most severe condition [17].

**TABS.**   TABS is a tool used by a child's caretaker to assess the current and past behaviors of children with autism. Autistic features change over the course of development; thus, for a clinician should know their behaviors in early infancy. It comprises 39 items that are provisionally grouped into four areas: interpersonal–social relationship, language–communication, habit–mannerism, and others. Each item is scored from 0 to 2 and has a highest score of 78. It is applied to both the general population and those diagnosed with ASD. It has no cut-off point. Higher scores indicate strong current autistic characteristics [15].

**Table 1. Clinical characteristics of the participants.**

|  | Case, % (n = 40) | Control, % (n = 40) | Effect size | p-value |
|---|---|---|---|---|
| Age (mean ± SD) | 8.55 ± 2.50 | 8.55 ± 2.50 | – | – |
| Male sex (%) | 87.5 (n = 35) | 87.5 (n = 35) | – | – |
| Antisocial behavior | 35.0 (n = 14) | 25.0 (n = 25) | – | NS |
| Child-to-parent violence | 27.5 (n = 11) | 2.5 (n = 1) | – | <0.01 |
| Suicide attempt or self-harm | 10.0 (n = 4) | None | – | NS |
| Refusal to attend school | 10.0 (n = 4) | 7.5 (n = 3) | – | NS |
| Abuse experience | 12.5 (n = 5) | 22.5 (n = 9) | – | NS |
| Abuse: physical | 7.5 (n = 3) | 15.0 (n = 6) | – | NS |
| Abuse: psychological | 7.5 (n = 3) | 10.0 (n = 4) | – | NS |
| Abuse: witnessing violence between parents | 5.0 (n = 2) | 5.0 (n = 2) | – | NS |
| Abuse: neglect | None | 5.0 (n = 2) | – | NS |
| Abuse: sexual | None | None | – | NS |
| TABS score | 24.43 ± 12.30 | 15.15 ± 7.00 | 0.93 | <0.01 |
| ADHD-RS total score | 33.18 ± 11.38 | 24.75 ± 8.69 | 0.83 | <0.01 |
| ADHD-RS hyperactivity/impulsiveness score | 14.35 ± 6.59 | 9.35 ± 5.41 | 0.83 | <0.01 |
| ADHD-RS inattention score | 18.83 ± 6.05 | 15.40 ± 5.34 | 0.60 | <0.01 |

Score ranges: Age, ≤15 years; TABS, 0–78 points; ADHD-RS, 0–54 points

ADHD-RS; attention-deficit hyperactivity disorder rating scale; NS, not significant; SD, standard deviation; TABS, Tokyo autistic behavior scale

## Statistical analysis

Pearson's chi-square test was used to compare proportions of binary variables between two groups, and Mann–Whitney U test was used to compare continuous variables between two groups (Table 1). Multivariate regression analysis was performed to determine variables that were independently associated with patients who received three ADHD medications (OROS methylphenidate, atomoxetine, and guanfacine). Possible confounding factors selected from previous studies and medical viewpoints, as described in the Introduction section, were included for multivariate logistic analysis. Odds ratios and 95% confidence interval were calculated using univariate and multivariate logistic regression models (Tables 2 and 3). Analyses with a p-value of <0.05, as shown in Tables 1 and 2, were included in the multivariate logistic regression analysis (Table 3). Appropriate variables were selected to prevent overfitting and multicollinearity. All statistical tests were two-tailed, and p-values of <0.05 were considered statistically significant. Analyses were performed using the Easy R Package version 1.40 [18].

**Table 2. Univariate logistic regression analyses.**

|  | OR | 95% CI | p-value |
|---|---|---|---|
| Antisocial behavior | 1.62 | 0.61–4.25 | NS |
| Child-to-parent violence | 14.8 | 1.81–121 | <0.01 |
| Refusal to attend school | 1.37 | 0.29–6.56 | NS |
| Abuse experience | 0.49 | 0.15–1.63 | NS |
| Abuse: physical | 0.46 | 0.11–1.98 | NS |
| Abuse: psychological | 0.73 | 0.15–3.49 | NS |
| Abuse: witnessing violence between parents | 1.00 | 0.13–7.47 | NS |

CI, confidence interval; NS, not significant; OR, odds ratio

**Table 3. Multivariate logistic regression analyses.**

|  | Multivariate adjusted OR | 95% CI | P-value |
|---|---|---|---|
| **Child-to-parent violence** | 10.9 | 1.22–96.9 | <0.05 |
| **TABS score** | 1.08 | 1.00–1.16 | <0.05 |
| **ADHD-RS total score** | 1.06 | 1.00–1.13 | <0.05 |

Note:

Each parameter was adjusted for Child-to-parent violence,

TABS score, and ADHD-RS total score

CI, confidence interval; OR, odds ratio

## Results

### Prescribing status of ADHD medications

Among the 3,900 patients who visited our department, 878 (22.5%) were prescribed ADHD medications. Of these 878 patients, 642 (73.1%) used OROS methylphenidate, 267 (30.4%) used atomoxetine, and 262 (29.8%) used guanfacine. Forty three of the 878 (4.9%) patients received OROS methylphenidate, atomoxetine, and guanfacine at least once (Fig 1). The proportion of patients prescribed a single medication only for ≥180 days were as follows: 75.6% for OROS methylphenidate; 68.3% for atomoxetine; and 55.3% for guanfacine.

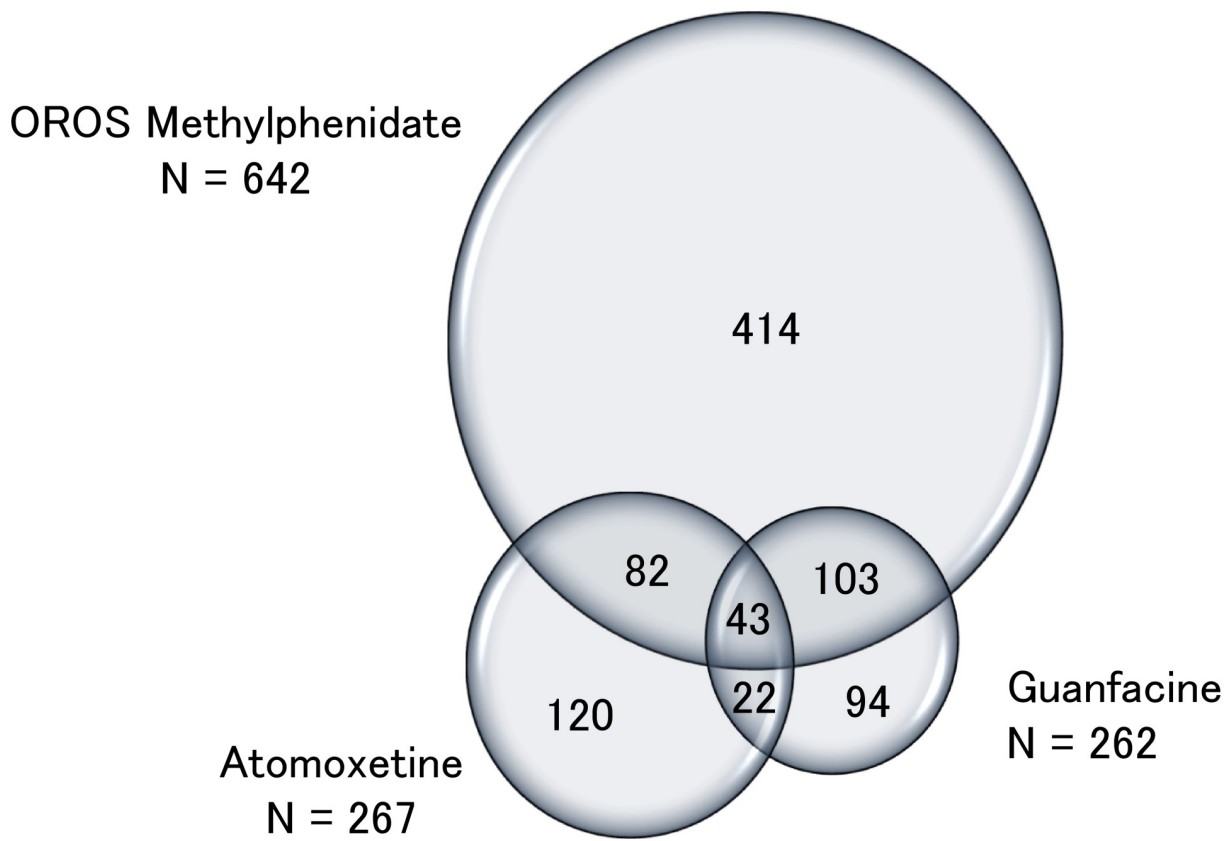

**Fig 1. Prescribing status of ADHD attention-deficit hyperactivity disorder medications.**

### Clinical characteristics of the participants

Selection of the case and control groups is described in Figs 2 and 3. Clinical characteristics of the participants are presented in Table 1. TABS, ADHD-RS total, ADHD-RS hyperactivity/impulsiveness, ADHD-RS inattention score, and proportion of child-to-parent violence were significantly higher in the case group than in the control group.

Logistic regression analyses revealed that TABS, ADHD-RS total score, and proportion of child-to-parent violence were independently associated with patients who received three medications, OROS methylphenidate, atomoxetine, and guanfacine, (case group) after adjusting for three parameters (Tables 2 and 3).

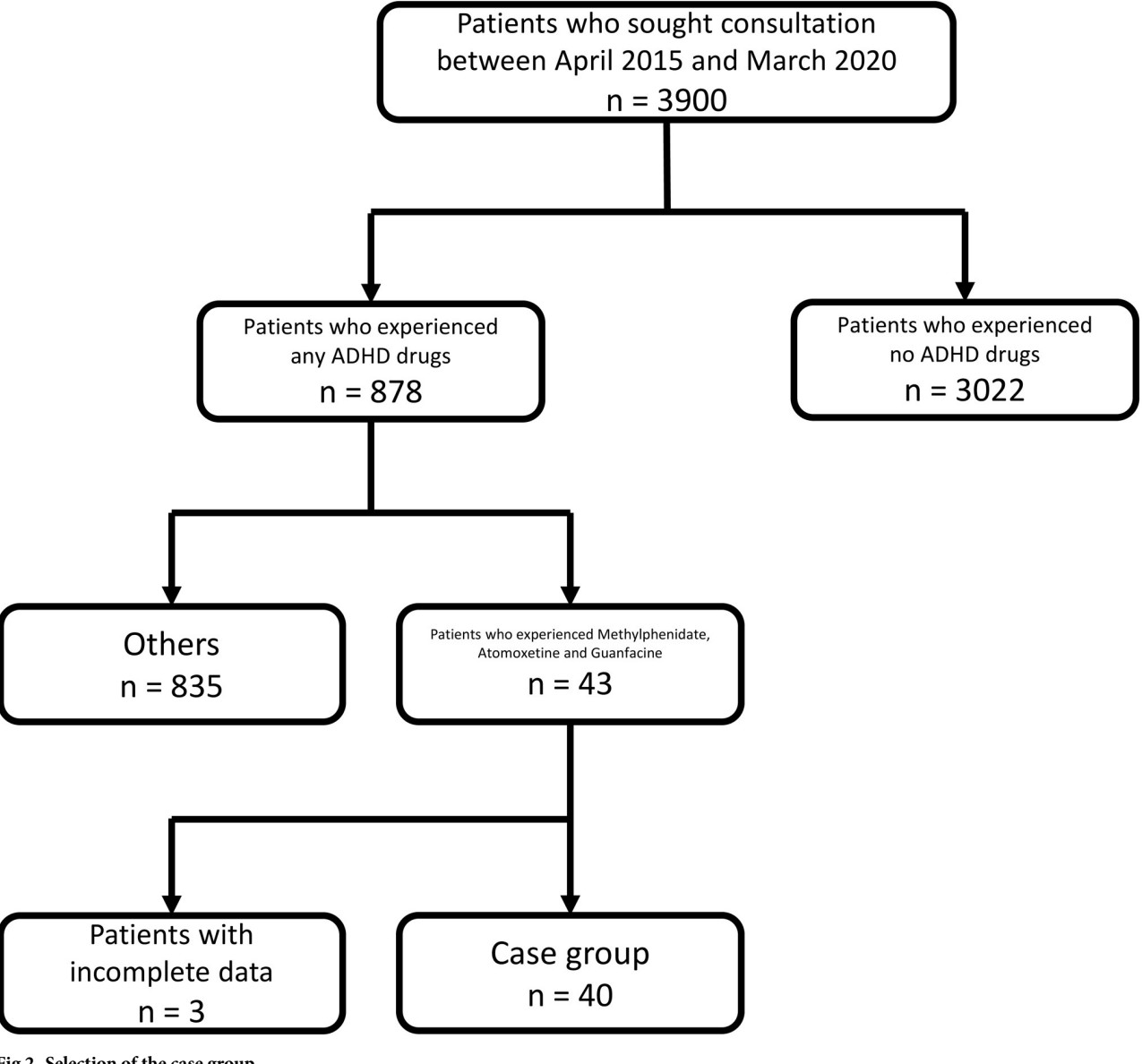

**Fig 2. Selection of the case group.**

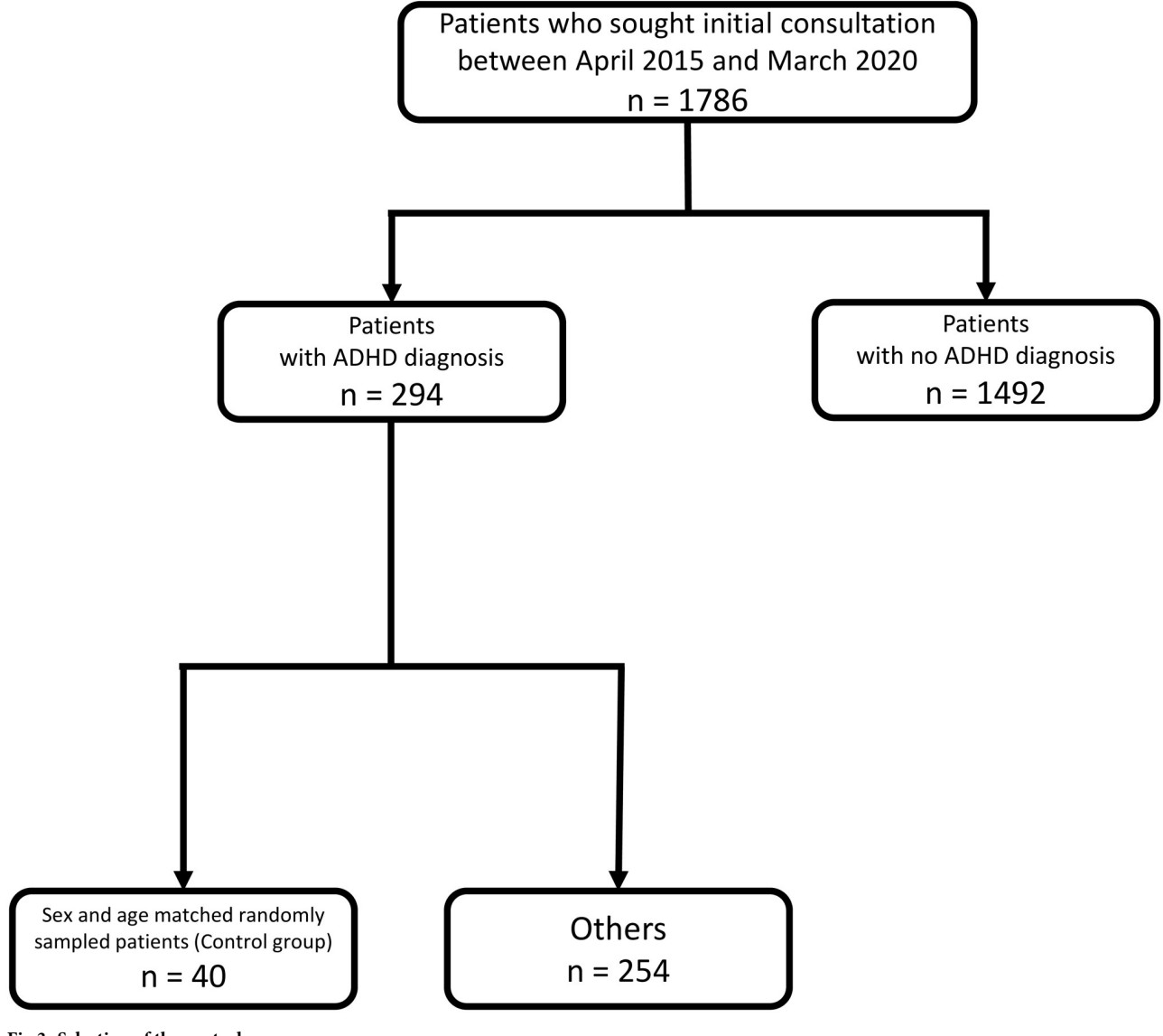

**Fig 3. Selection of the control group.**

## Discussion

This is the first study to evaluate the clinical characteristics of child and adolescent psychiatric outpatients prescribed three medications (OROS methylphenidate, atomoxetine, and guanfacine) in Japan.

OROS methylphenidate was the most frequently prescribed medication in this study. This result was consistent with a retrospective cohort study using the National Database of Health Insurance Claim Information, in which the percentage of methylphenidate use among ADHD medication users was 64% in Japan [19]. Considering that methylphenidate is the most frequently used medication in children with ADHD in the UK (94%; [20]), Norway (94%; [21]), and Germany (75%–100%; [22]), the finding of 73.1% in this study is not high. This may be partly explained by the restriction policy in Japan that applies to OROS methylphenidate. The

fact that immediate-release methylphenidate has not been approved for ADHD treatment in Japan may have also contributed to the low prevalence of methylphenidate prescriptions.

Among the 878 patients who were prescribed ADHD medications, 43 (4.9%) received three medications. The response rate to a specific stimulant (reduction in hyperactivity or increase in attention) was approximately 70% [23, 24]. At least 50% of children who did not respond to one type of stimulant responded to another [25, 26]. Of the 30% of patients who did not respond to the first ADHD medication, half of them did not respond to the second medications. So, the estimated percentage of users of three medications was approximately 15% based on these previous findings. The difference may be explained by the fact that this study was conducted from April 2015 to March 2020, and guanfacine became available in May 2017.

The clinical characteristics associated with the use of three medications were severe ADHD symptoms, autistic characteristics, or a tendency of child-to-parent violence at the initial consultation. The choice of ADHD medications for severe ADHD symptoms and associated autistic characteristics may still be difficult. Functioning of ADHD in early adulthood was predicted by persistence of symptoms, baseline ADHD severity, intelligence quotient, and comorbidity [27, 28]. There is not enough evidence that ADHD medications improve social interaction, stereotypical behavior, or overall autistic characteristics [29]. In other words, autistic characteristics need to be assessed when ADHD medications are ineffective and additional therapeutic approaches are required. Conversely, it may suggest that ADHD medications are often prescribed for comorbid autistic characteristics because it is difficult to distinguish ADHD symptoms from autistic characteristics [30]. Regarding child-to-parent violence, ADHD medications might be prescribed to reduce impulsivity as much as possible because reactive-impulsive violence was related to ADHD psychopathology [31]. Cohort studies reported that depression in young people was associated with subsequent violent outcomes [32]. Furthermore, a systematic review and meta-analysis reported that child maltreatment or exposure to domestic violence was associated with an increased risk of interpersonal violence [33]. This finding suggests the need for future research on the relationship between patients with child-to-parent violence who receive multiple medications and a history of depression and abuse.

Another reason for patients requiring three medications may be the low prescription rates of OROS methylphenidate in Japan. The proportion of patients prescribed a single drug only for ≥180 days were 75.6% for OROS methylphenidate and 68.3% for atomoxetine. According to a previous study, OROS methylphenidate showed a higher persistency rate than atomoxetine for patients with ADHD or pervasive developmental disorder with symptoms of ADHD in a Japanese clinical setting [34]. Because OROS methylphenidate had a greater effect size for ADHD symptoms than atomoxetine and guanfacine [35], the initial use of ADHD medications with a smaller effect size may have led to the use of multiple ADHD medications. In addition, Japanese child and adolescent psychiatrists may be more cautious about the risks than the benefits owing to concerns that OROS methylphenidate may have adverse effects on growth and lead to dependence and abuse [6, 36]. Further studies on medication selection for ADHD are required.

## Study limitations

This study has some limitations that should be considered. First, a selection bias may exist owing to several reasons. We did not track drug information before our consultation; therefore, patients in the control group may have been prescribed multiple ADHD medications. Furthermore, this study did not represent the general situation associated with use multiple ADHD medications among child and adolescent psychiatric patients because it was conducted

in a single district by a small number of psychiatrists. Second, this study may have been affected by information bias owing to several reasons. Information on a diagnosis was established after the first consultation. However, a diagnosis may change, and additional information may be included after further examinations. In total, 878 participants had a diagnosis of ADHD but comorbidities such as ASD were not examined. Information on the dosage and side effects of each ADHD medication, in what order and for how long it was prescribed, was not evaluated between two groups. Distinguishing whether ADHD medications were used simultaneously or sequentially would reveal more acute clinical characteristics of patients who are resistant to ADHD medications. Information on abuse history was established after the first consultation. However, additional information regarding abuse history may be included after further examinations. Inaccuracies were present in the detection of abuse history because many abuses were not immediately visible. In other words, we might have missed an abuse history. In addition, this study did not include information on factors other than developmental characteristics and abuse, such as depression, and use of atypical antipsychotics for autistic characteristics. Because we wanted to make the study more relevant to actual clinical situations that fits the clinician's sense, we did not divide the transition period in May 2017 when guanfacine became available. The results might change if the study was conducted after guanfacine is widely available. Therefore, this study only provides a partial view of the clinical characteristics of the patients received multiple ADHD medications. Finally, the results of this study confirmed the association between use of three medications and clinical characteristics, but not causality.

## Conclusion

This study examined the prescribing status of ADHD medications in a Japanese clinical setting and the clinical characteristics of child and adolescent psychiatric patients who received three medications (OROS methylphenidate, atomoxetine, and guanfacine) at least once. Among the 878 patients who were prescribed any ADHD medications, 43 (4.9%) patients received OROS methylphenidate, atomoxetine, and guanfacine. Logistic regression revealed that children with severe ADHD symptoms, autistic characteristics, or with a tendency of child-to-parent violence at the initial consultation were more likely to be prescribed three medications during the course of treatment. It is important to consider that these findings from the initial consultation are important to prevent the use of multiple ADHD medications.

## Acknowledgments

The authors would like to thank Enago (www.enago.jp) for the English language review.

## Author Contributions

**Conceptualization:** Yoshinori Sasaki, Noa Tsujii, Takayuki Okada, Masahide Usami.

**Data curation:** Yoshinori Sasaki, Shouko Sasaki, Hikaru Sunakawa, Yusuke Toguchi, Syuuichi Tanase, Kiyoshi Saito, Rena Shinohara, Toshinari Kurokouchi, Kaori Sugimoto, Kotoe Itagaki, Yukino Yoshida, Saori Namekata, Momoka Takahashi, Ikuhiro Harada, Yuuki Hakosima, Kumi Inazaki, Yuta Yoshimura, Yuki Mizumoto, Masahide Usami.

**Formal analysis:** Yoshinori Sasaki, Takayuki Okada, Masahide Usami.

**Funding acquisition:** Masahide Usami.

**Investigation:** Yoshinori Sasaki, Noa Tsujii, Shouko Sasaki, Hikaru Sunakawa, Yusuke Toguchi, Syuuichi Tanase, Kiyoshi Saito, Rena Shinohara, Toshinari Kurokouchi, Kaori

Sugimoto, Kotoe Itagaki, Yukino Yoshida, Saori Namekata, Yuuki Hakosima, Kumi Inazaki, Yuta Yoshimura, Yuki Mizumoto, Takayuki Okada, Masahide Usami.

**Methodology:** Yoshinori Sasaki, Noa Tsujii, Shouko Sasaki, Hikaru Sunakawa, Yusuke Toguchi, Syuuichi Tanase, Kiyoshi Saito, Rena Shinohara, Toshinari Kurokouchi, Kaori Sugimoto, Kotoe Itagaki, Yukino Yoshida, Saori Namekata, Momoka Takahashi, Ikuhiro Harada, Yuuki Hakosima, Kumi Inazaki, Yuta Yoshimura, Yuki Mizumoto, Takayuki Okada, Masahide Usami.

**Project administration:** Takayuki Okada, Masahide Usami.

**Supervision:** Noa Tsujii, Takayuki Okada, Masahide Usami.

**Validation:** Yoshinori Sasaki.

**Writing – original draft:** Yoshinori Sasaki.

**Writing – review & editing:** Yoshinori Sasaki, Noa Tsujii, Takayuki Okada, Masahide Usami.

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
