## [Decision Letter · Decision Letter 0]

19 Mar 2021

PONE-D-21-04661

Status of attention-deficit hyperactivity disorder (ADHD) drugs and clinical characteristics of child and adolescent psychiatric outpatients prescribed multiple ADHD drugs in Japan

PLOS ONE

Dear Dr. Usami,

Thank you for submitting your manuscript to PLOS ONE. After careful consideration, we feel that it has merit but does not fully meet PLOS ONE’s publication criteria as it currently stands. Therefore, we invite you to submit a revised version of the manuscript that addresses the points raised during the review process.

The three reviewers addressed a number of major and minor concerns about your manuscript. Please revise your manuscript carefully.

We look forward to receiving your revised manuscript.

Kind regards,

Kenji Hashimoto, PhD

Academic Editor

PLOS ONE

Journal Requirements:

Reviewers' comments:

Reviewer's Responses to Questions

**Comments to the Author**

1. Is the manuscript technically sound, and do the data support the conclusions?

Reviewer #1: Partly

Reviewer #2: Partly

Reviewer #3: Yes

2. Has the statistical analysis been performed appropriately and rigorously? 

Reviewer #1: N/A

Reviewer #2: N/A

Reviewer #3: Yes

3. Have the authors made all data underlying the findings in their manuscript fully available?

Reviewer #1: Yes

Reviewer #2: No

Reviewer #3: Yes

4. Is the manuscript presented in an intelligible fashion and written in standard English?

Reviewer #1: Yes

Reviewer #2: Yes

Reviewer #3: No

5. Review Comments to the Author

Reviewer #1: The authors investigated pediatric ADHD patients who experienced osmotic-controlled release oral delivery system methylphenidate, atomoxetine, or guanfacine. A retrospective case–control design was used to evaluate the characteristics of patients who experienced multiple ADHD drugs.

The authors stated that logistic regression revealed that children with severe ADHD symptoms, autistic characteristics, or tendency of child-to-parent violence were more likely to have been prescribed three drugs during their treatment.

The strength of this study is the first study to evaluate the clinical characteristics of child and adolescent psychiatric outpatients prescribed three drugs (OROS methylphenidate, atomoxetine, and guanfacine) in Japan.

However I have a few questions.

#1 The authors stated that the clinical characteristics associated with the use of three drugs were severe ADHD symptoms, autistic characteristics, or a tendency of child-to-parent violence at the initial consultation.

How do you assess these autistic characteristics, or a tendency of child-to-parent violence?

I think that these symptoms might be caused by the symptoms of Post Traumatic Stress Disorder.

#2 Related to #1, How do you assess the abuse types?

Reviewer #2: Thank you very much for giving me an opportunity to review this article.

Because the reason to choose certain prescription pattern is not clear, such kind of data finding is interesting for many clinicians. The sample size is large enough.

However, the following points weaken the results:

1) This study is done only in medical facility. Then, the result only reflects prescription pattern of small number of psychiatrists.

2) Medical information (such as the history of being abused) are based on only medical records.

3) Samples are collected in 2015-2020. Among this period, some new anti-ADHD agents are marketed.This changed the prescription status. Authors must address why authors can collapse these data into the same datasets.

In addition, authors should examine the following points.

The PDF of Registry Study of this article is written in Japanese. It seems an opt-out.

Authors must change tables and figures. They are unclear.

Reviewer #3: It was a pleasure to go through your valuable work, examining the current trend in ADHD medication use in Japan and factors related to multidrug therapy in children. I would like to clarify a few points for further improvement in the presentation of your material.

1. Please reconsider the title of the manuscript, since it is somehow ambiguous.

2. In the Abstract at the Conclusion section, you stated that your findings highlighted the approach to prevent multiple medications in children with ADHD. I am afraid that while you talked about the characteristics and possible reasons behind the multiple medications, I could not find notions regarding the approach/ways to avoid the multiple medications in the Discussion.

3. The term prescription status sounds confusing. It refers to a status of a patient prescription (either active, pending, or cancelled etc.). Similarly, prescribing status means if a physician or pharmacist is currently prescribing medications or not. I think you are discussing the current medication use and it would be better to re-check the use of those terms with the English editing service. Moreover, could you also check the term patients who experienced multiple ADHD drugs? I reckon patients who were multiply medicated or who received multiple ADHD medications or patients who were treated with all three ADHD medications is more appropriate. The word experience highlights the meaning of subjective experience of a person taking medication.

4. In the final paragraph of the Introduction, you state that although it is not appropriate to simultaneously prescribe all three ADHD medications, a sequential trial of those is reasonable. You had better clarify why it is not appropriate to simultaneously prescribe all three ADHD medications.

5. In the final paragraph of the Introduction, it is better to present the argument of ADHD-ASD co-occurrence/ADHD medication use in ASD (i.e. reasons behind your hypothesis) before stating your hypothesis.

6. In the Method, the inclusion and exclusion criteria are not presented clearly enough. What was the targeted age range (the minimum age)? Did you include patients who were on any of the three ADHD medications continuously for more than 180 days regardless of their diagnoses? What were the diagnoses of 878 participants including comorbidity? As you mentioned in the Introduction, ADHD and ASD co-occur to a significant degree. Did you exclude or include (for the whole sample and the case and control groups) individuals with ASD and if so how did you make the diagnosis (like using the ADOS-2 or ADI-R)?

Further, could you describe what you mean by autistic characteristics (Does this mean patients presenting some ASD symptoms but without ASD diagnosis or include those with ASD diagnosis)? In addition, how did you diagnose ADHD (I understand that expert psychiatrists and psychologists in the field established interview forms and made diagnoses but there are no references to the intake of developmental history from parents, observation of a child, or reviewing of school records etc.)? More details for the description of the diagnostic process are preferable.

7. You should describe the medication dosage of the two groups.

8. For the two-group comparison, I would like to clarify that it is comparing ADHD diagnosed children with a use of no more than 2 ADHD drugs (i.e. tried zero or one or two drugs but not all three) versus ADHD diagnosed children who were treated with all three ADHD medications. Were they diagnosed only with ADHD without any comorbidity including ASD/depression/anxiety disorders?

9. For those who used more than 2 medications, did they switch medications due to insufficient treatment effects or was the switch due to adverse effects?

10. You excluded patients with moderate to severe intellectual disability. How did you assess the intelligence?

11. How did you assess the child-to-parent violence, anti-social behaviors, or abuse? Was it based on a parent report or asked to a patient in the clinician interview? It is possible that reports on these sensitive episodes, especially abuse, are different between those made by parents and those by children and I am wondering how and from whom you extracted this information.

12. Is TABS applied to a general population to assess ASD symptoms like the Autism Spectrum Quotient or only to assess current ASD symptoms in those diagnosed with ASD? Is there any cut-off? It may be helpful to explain it in detail as it is used only in Japan.

13. In the Method about the statistical analysis, the sentences for deciding on the explanatory variables for the univariate and multivariate analyses are not clear.

14. In the Result, the age ranged from 10 to 15 years old (written below Table1), however the mean age is 8.55. How is this possible?

15. In the Discussion Page 9 Line 19, approximately is not spelled right.

16. In the Discussion Page 10 Line 12-18, you talk about the relationship between depression and violence and its possibility underlying the insufficient response to a single ADHD medication. I understand the importance but did you assess the depression/anxiety in the study or did you include those with ADHD comorbid with depression? Did any of the participants used anti-depressives along with ADHD medications? It seems a little leap to state the association with depression without mentioning such information.

6. PLOS authors have the option to publish the peer review history of their article (what does this mean?). If published, this will include your full peer review and any attached files.

Reviewer #1: No

Reviewer #2: No

Reviewer #3: No

---

## [Author Response · Author response to Decision Letter 0]

23 Apr 2021

Thank you for reviewing our manuscript and providing us your valuable advice despite your busy schedules.

We addressed all points raised by the academic editor and reviewers. In addition, to better facilitate the readers’ understanding, we modified some parts of the manuscript that were not pointed out by the reviewers. After these changes, scientific and academic editing services were used to re-edit the revised version of our manuscript before resubmission.

We would appreciate it if you could let us know again whether there are still parts that need further explanation or should be corrected.

---

## [Decision Letter · Decision Letter 1]

17 May 2021

Status of attention-deficit hyperactivity disorder (ADHD) drugs and clinical characteristics of child and adolescent psychiatric outpatients prescribed multiple ADHD drugs in Japan

PONE-D-21-04661R1

Dear Dr. Usami,

We’re pleased to inform you that your manuscript has been judged scientifically suitable for publication and will be formally accepted for publication once it meets all outstanding technical requirements.

Kind regards,

Kenji Hashimoto, PhD

Section Editor

PLOS ONE

Additional Editor Comments (optional):

Reviewers' comments:

Reviewer's Responses to Questions

**Comments to the Author**

1. If the authors have adequately addressed your comments raised in a previous round of review and you feel that this manuscript is now acceptable for publication, you may indicate that here to bypass the “Comments to the Author” section, enter your conflict of interest statement in the “Confidential to Editor” section, and submit your "Accept" recommendation.

Reviewer #1: All comments have been addressed

Reviewer #2: (No Response)

Reviewer #3: (No Response)

2. Is the manuscript technically sound, and do the data support the conclusions?

Reviewer #1: Yes

Reviewer #2: No

Reviewer #3: Yes

3. Has the statistical analysis been performed appropriately and rigorously? 

Reviewer #1: I Don't Know

Reviewer #2: No

Reviewer #3: Yes

4. Have the authors made all data underlying the findings in their manuscript fully available?

Reviewer #1: Yes

Reviewer #2: No

Reviewer #3: Yes

5. Is the manuscript presented in an intelligible fashion and written in standard English?

Reviewer #1: Yes

Reviewer #2: Yes

Reviewer #3: Yes

6. Review Comments to the Author

Reviewer #1: Thank you for your polite reply.

Response to Reviewer #1

I have a few questions. #1 The authors stated that the clinical characteristics associated with the use of three drugs were severe ADHD symptoms, autistic characteristics, or a tendency of child-to-parent violence at the initial consultation. How do you assess these autistic characteristics, or a tendency of child-to-parent violence? I think that these symptoms might be caused by the symptoms of Post Traumatic Stress Disorder. #2 Related to #1, How do you assess the abuse types?

As per Reviewer #1’s comments, we added a description in the Recruitment and participants. Readers’ misunderstandings or questions can be avoided by explaining in detail what we have conducted at our facility.

REVISED

Psychologists and psychiatrists together established the initial interview forms, which included demographic and clinical characteristics. Consultation included the developmental history from caretakers, child observations, and school record reviews. TABS and ADHD-RS were then constructed. Psychiatrists specializing in child and adolescent psychiatry evaluated clinical findings, including psychological rating scales and diagnosed all patients according to the Diagnostic and Statistical Manual of Mental Disorders, Fifth Edition [1]. Meetings to discuss the psychopathology, symptoms, diagnosis, and possibility of abuse were conducted as needed. There were few patients Patients with moderate-to-severe intellectual disability according to the DSM-5, organic brain disease, drug-induced psychiatric disease, traumatic brain injury, and genetic syndromes because they were referred to other medical institutions when they call for an appointment in and were excluded from this study.

In this study, child the TABS was filled out by caretakers and then psychiatrists checked the items in a medical interview. The TABS score indicated the strength of autistic characteristics, and the ADHS-RS indicated the strength of ADHD symptoms as well. Child-to-parent violence meant physical violence against parents living together or damage to property. The in-charge psychiatrists interviewed patients and parents individually, as needed, to evaluate for child-to-parent violence. After hearing both sides of the story, psychiatrists decided whether the patients’ behavior was indicative of child-to-parent violence. We considered patients as having “antisocial behavior” when patients engaged in illegal activities, such as smoking, drinking, illegal drug use, shoplifting, or causing injury to others. In Japan, smoking and drinking are legal after the age of 20 years. In many cases, the evaluation of antisocial behavior would be based on information from parents. “Abuse experience” included sexual abuse, physical abuse, psychological abuse, and neglect. Our department worked with local child protection services. Hence, information about abused children was often available before the initial consultation. In addition, the in-charge psychiatrists interviewed parents and patients separately to identify abused children as needed. These efforts were made to immediately identify abused children and to prevent missed abused children. Refusal to attend school was defined as absence from school for ≥30 days owing to any psychological, emotional, physical, or social reasons [16].

Reviewer #2: Authors agree to my comments, however, they only add description to limitations. These points substantially lost the significance of this article. Therefore, I think that this article does not reach the standard for publication of PLOS ONE.

Reviewer #3: (No Response)

7. PLOS authors have the option to publish the peer review history of their article (what does this mean?). If published, this will include your full peer review and any attached files.

Reviewer #1: No

Reviewer #2: No

Reviewer #3: No

---

## [Editor Report · Acceptance letter]

24 May 2021

PONE-D-21-04661R1 

Current use of attention-deficit hyperactivity disorder (ADHD) medications and clinical characteristics of child and adolescent psychiatric outpatients prescribed multiple ADHD medications in Japan 

Dear Dr. Usami:

I'm pleased to inform you that your manuscript has been deemed suitable for publication in PLOS ONE. Congratulations! Your manuscript is now with our production department. 

Kind regards, 

on behalf of

Prof. Kenji Hashimoto 

Section Editor

PLOS ONE